# Regulatory Effects and Interactions of the Wnt and OPG-RANKL-RANK Signaling at the Bone-Cartilage Interface in Osteoarthritis

**DOI:** 10.3390/ijms20184653

**Published:** 2019-09-19

**Authors:** Béla Kovács, Enikő Vajda, Előd Ernő Nagy

**Affiliations:** Department of Biochemistry and Environmental Chemistry, University of Medicine, Pharmacy, Sciences and Technology, Tîrgu Mureș, Romania; bela.kovacs@umfst.ro (B.K.); cs_encike@yahoo.com (E.V.)

**Keywords:** osteoarthritis, Wnt signaling, bone-cartilage interface

## Abstract

Cartilage and the bordering subchondral bone form a functionally active regulatory interface with a prominent role in osteoarthritis pathways. The Wnt and the OPG-RANKL-RANK signaling systems, as key mediators, interact in subchondral bone remodeling. Osteoarthritic osteoblasts polarize into two distinct phenotypes: a low secretory and an activated, pro-inflammatory and anti-resorptive subclass producing high quantities of IL-6, PGE2, and osteoprotegerin, but low levels of RANKL, thus acting as putative effectors of subchondral bone sclerosis. Wnt agonists, Wnt5a, Wisp-1 initiate excessive bone remodeling, while Wnt3a and 5a simultaneously cause loss of proteoglycans and phenotype shift in chondrocytes, with decreased expression of COL2A, aggrecan, and Sox-9. Sclerostin, a Wnt antagonist possesses a protective effect for the cartilage, while DKK-1 inhibits VEGF, suspending neoangiogenesis in the subchondral bone. Experimental conditions mimicking abnormal mechanical load, the pro-inflammatory milieu, but also a decreased OPG/RANKL ratio in the cartilage, trigger chondrocyte apoptosis and loss of the matrix via degradative matrix metalloproteinases, like MMP-13 or MMP-9. Hypoxia, an important cofactor exerts a dual role, promoting matrix synthesis via HIF-1α, a Wnt silencer, but turning on HIF-2α that enhances VEGF and MMP-13, along with aberrant collagen expression and extracellular matrix deterioration in the presence of pro-inflammatory cytokines.

## 1. Introduction

Osteoarthritis is a complex, chronic disease that affects primarily the weight-bearing joints of humans and other mammals. During progression of the disease, all compartments of the joints undergo structural, functional and metabolic changes that involve cellular elements as well as components of the extracellular matrix. Classically, degenerative deterioration of the cartilage was thought to be the central process of osteoarthritis. However, the results of recent studies reveal that deterioration of the cartilage can be preceded by synovitis, low-grade inflammation, and remodeling of the subchondral bone [1]. The bone-cartilage interface is an important anatomic region, which plays a crucial role in the degradative pathways of osteoarthritis [2], but the interactions of the cartilage and subchondral bone are not well understood. This review focuses on the regulatory processes at the cartilage-subchondral bone interface that may transform a functioning joint to one with osteoarthritis.

## 2. Structural and Functional Deterioration of the Cartilage are Central Processes in Osteoarthritis

Chondrocytes, the only cells of the cartilage, are responsible for the biochemical turnover, synthesis, and secretion of the extracellular matrix component; the cells differ in shape and size in various areas of cartilage, but all forms contain cellular organelles required for matrix synthesis. Chondrocytes adapt their metabolism to physicochemical changes of the microenvironment as mechanical and osmotic sensors [3,4]. Their density is the highest in the deep cartilage layer and progressively decreases in the middle transitional and superficial area to about 1/3 of the density in the deep zone. Individually, chondrocytes have a high metabolic turnover, but their overall activity determines slow changes in the cartilage due to the low cell density (chondrocytes account for less than 5% of the total cartilage tissue volume).

The extracellular matrix of articular cartilage consists of a basic amorphous, gelatinous substance and collagen fibers. The matrix consists of 65–80% water and inorganic salts and 20–35% organic macromolecular components: collagen, proteoglycans (aggrecan), other proteins, and glycoproteins [4]. Collagen fibers form a three-dimensional network that provides the tissue shape, immobilizes proteoglycans, limits their hydration, and gives cartilage mechanical resistance. Type II collagen is specific for articular cartilage and is the dominant fiber component (90–98%) together with collagen types VI, IX, X, XI [4]. Covalently cross-linked peptides of type II collagen are secreted and can be used as biomarkers of cartilage degradation both in humans and experimental animals [5,6,7].

Cartilage degeneration involves degradation of the extracellular matrix due to action of catabolic enzymes and apoptosis of chondrocytes. Alteration of the collagen network and the loss of proteoglycans ultimately lead to visible destruction of the extracellular matrix [8]. At the microscopic level, the destruction manifests as focal matrix condensation, edema, fissures and fibrillation, local erosions, and cartilage denudation along with formation of osteophytes. Alteration of the extracellular matrix causes deficits in biomechanical function. The relaxation of collagen fibers leads to loss of aggrecan and to mechanical overload, thus weakening the collagen network. Degradative processes are especially characteristic at the surface of the cartilage and around the chondrocytes [8].

The enzymes responsible for collagen degradation are collagenases secreted by chondrocytes, called matrix metalloproteinases (MMPs). Cleavage of aggrecan is mediated by aggrecanases (especially ADAMTS-4 and ADAMTS-5, ADAMTS—a disintegrin and metalloproteinase with thrombospondin motifs) [9]. Low-grade inflammation in the subchondral bone and bone marrow have been reported both in experimental models and human arthritis [10,11]. In SW1353 chondrosarcoma cells, IL-1β (interleukin-1β) induces overexpression of IL-6 (interleukin-6), PGE2 (prostaglandin E2) and Cox-2 (cyclooxygenase 2), which, in turn, activate a series of MMPs, such as MMP-1 and MMP-13 [12]. Thus, local inflammatory signals in the osteoarthritic cartilage turn on degradative enzymes. Besides monocytes/macrophages infiltrating the synovial membrane, an important source of the pro-inflammatory mediators released to the cartilage, might be the subchondral bone, and, especially, subchondral osteoblasts.

## 3. Molecular Crosstalk through the Bone-Cartilage Interface

The characteristic changes of the subchondral bone are biphasic in their timeline. In the initial stages, the metabolic turnover increases, and the subchondral bone plate becomes thinner. In a later stage, the tidemark delimiting the calcified and non-calcified cartilage becomes discontinuous and gradually disappears; the bone plate thickens and becomes sclerotic, while the bone marrow loses volume and undergoes fibrotic transformation [1].

Calcified cartilage is not an impenetrable barrier; rather, it permits the diffusion of various-sized metabolites and regulatory molecules. Due to the physical proximity and the intimate interactions of the two histological regions, the abundant subchondral vasculature, the fissures, and the matrix pores developing in the calcified cartilage are newly formed communication channels for soluble molecules [13]. Some workers have reported that more than 50% of the glucose and oxygen requirement of the cartilage is supplied from the subchondral bone marrow [14]. Small molecules can diffuse from the bone marrow through the subchondral bone and deep layers of cartilage to the synovial fluid [15]. In a comparative study, the diffusion coefficients present in early human osteoarthritis between the various zones of cartilage and the subchondral bone plate were slightly higher than those in healthy equine cartilage layers, the ratios of diffusion coefficients being comparable [16]. The subchondral bone and the calcified cartilage in osteoarthritis undergo transformations that favor the transport of larger soluble molecules. As disease progresses, increased porosity in the extracellular matrix and active angiogenesis occur [17,18]. The biochemical crosstalk might be bidirectional, as cells of the subchondral bone and bone marrow seem to be a major source of secreted regulatory molecules in osteoarthritis. The appearance of the bone marrow lesions is an early phenomenon; they appear in close proximity to the covering cartilage lesions and generally are irreversible [19,20]. Furthermore, there is a correlation between the size of bone marrow and cartilage lesions. Little is known about the intimate histological changes occurring in the bone marrow. The most important scoring systems used to assess disease severity in experimental animals, such as the Osteoarthritis Research Society International Histopathology Initiative, apply either global structural, histomorphometric criteria (bone mineral density, measured by microcomputed tomography or dual energy x-ray absorptiometry, trabecular bone volume, and osteoid volume), or semi-quantitative appreciation of the histologic lesions (fragmentation of the tidemark, extent of bone marrow changes, and fibrosis in bone marrow) [21,22].

Bone marrow lesions are early magnetic resonance imaging diagnostic features of osteoarthritis and are closely correlated with the severity of pain. Their presence is a useful predictor of the rate of cartilage loss in patients with knee osteoarthritis [20,23]. These features, which are spatially associated with the overlying cartilage lesions, are present in 70% of explanted tibial plateaus of knee osteoarthritis patients, and they are characterized by bone-marrow edema, fibrosis, necrosis, and fibrovascular cyst-like formations [23].

## 4. The OPG-RANKL-RANK Regulatory System

Two research groups discovered osteoprotegerin (OPG) independently more than two decades ago. While investigating tumor necrosis factor-related genes and human embryonic fibroblast-secreted molecules, the workers discovered cDNA sequences coding a master regulator protein of osteoclastogenesis [24,25]. Applying the molecule as a probe in ligand-binding experiments, they also identified its ligand and initially called it OPG ligand (OPGL), then osteoclast differentiation factor, and later, receptor activator of nuclear kappa B ligand (RANKL) [26,27]. OPG is expressed in many tissue types, and its circulating levels are associated with endothelial dysfunction, coronary artery disease, and peripheral atherosclerosis [28,29,30]. RANKL is a homotrimeric transmembrane protein secreted by osteoblasts but also by immune and tumor cells, which stimulate in the bone the differentiation of osteoclasts and the release of immature progenitor cells into the circulation. RANKL is secreted through cleavage mediated by a disintegrin and metalloproteinase domain (ADAM) and MMP-7 [31,32]. The RANK receptor is a widely expressed homotrimeric molecule that has partial homology with the extracellular domain of CD40, an important co-activator of T lymphocytes [33]. At least seven downstream signaling pathways for RANK have been described; among these, three trigger osteoclastogenesis (nuclear factor kappa-light-chain-enhancer of activated B cells—NFκB, c-Jun amino-terminal kinase/activator protein-1, nuclear factor of activated T cells—NFAT-1) and others mediate osteoclast activation (Src and MKK6/p38/MITF). NFκB, in cooperation with c-fos, probably generates the strongest osteoclastogenic effect. Osteoblasts expressing RANKL control the differentiation of osteoclasts through RANK signaling. In the triad, OPG acts as a decoy receptor, blocking the effect of RANKL, thus suspending differentiation and activation of bone-resorbing cells [34]. In osteoblasts, OPG expression is regulated by the Wnt/β-catenin regulatory system [35]. OPG, RANKL and RANK are also expressed in cartilage, but their role in chondrocyte functions is poorly understood.

## 5. The Wnt Signaling Pathways—A Brief Overview

In the activation state, when cWnt glycoprotein ligands are bound, co-activation of the Frizzled (FZD) and LRP5/6 receptors by Wnt1 and Wnt7, for example, contribute to formation of a heterodimer between the receptors [36,37]. The formation of this complex in turn results in the degradation of the β-catenin destruction complex, leading to elevated intracellular β-catenin concentrations. Later, the protein is translocated into the nucleus, where it is a co-activator for various factors, e.g., T-cell factor (TCF) or lymphoid enhancer binding factor (LEF). Binding of TCF/LEF to DNA through the Wnt-responsive element provides gene activation by Groucho delocalization from the target gene and via recruitment of alternative co-factors [38,39,40,41]. Auxiliary pathways (Wnt/Planar Cell Polarity, Wnt/Ca^2+^ pathways) include the activation of FZD receptors and co-receptors, such as receptor tyrosine kinase-like orphan receptor and receptor-like tyrosine kinase, by ncWnt ligands, e.g., Wnt5a. This activity results in gene transcription via G-protein-coupled receptor-mediated intracellular communications or mitogen-activated protein kinase (MAPK) signaling pathways [41,42,43,44,45].

In oxidative stress, the canonical pathway is regulated by Forkhead box O (FOXO), a transcription factor that competes with TCF for β-catenin binding. FOXO silences TCF transcriptional activity and is an adapter to hypoxia [46].

## 6. Runt-Related Transcription Factor 2 is a Regulatory Link between the OPG-RANKL-RANK and the Wnt/ β-Catenin System

Osteoblasts are key actors in anabolic processes in bone. These cells synthesize the most important proteins of the extracellular matrix: collagen type I, bone sialoprotein, osteocalcin, and osteopontin. Osteoblasts also are responsible for mineralization of the matrix by deposition of calcium ions and phosphates. The cells control the differentiation of osteoclasts through the secretion of RANKL, which binds to a cell-surface receptor of osteoclasts, RANK. RANK sequentially induces TRAF6 (TNF receptor associated factor), MAPKs (mitogen-activated protein kinase), like JNK (c-jun N-terminal kinase), ERKs (extracellular signal-regulated kinase) and p38 and various transcription factors, such as canonical and non-canonical NFκB, c-fos and c-jun [47,48].

Osteoarthritic osteoblasts possess distinct functional features and molecular profile. The cells differ from normal osteoblasts and behave oppositely to the osteoporotic counterparts: they have higher OPG expression and lower RANKL expression, along with up-regulated metabolic activity, which is translated by high Runt-related transcription factor 2 (RunX2) and osterix levels, increased osteocalcin synthesis and alkaline phosphatase activity [49]. RunX2 expression is mandatory for bone formation, as in RunX2 -/- mice the cessation of bone formation occurs, due to maturational arrest of osteoblasts and lack of osteoclast differentiation and numeric reduction. However, RunX2 modulation is probably time-dependent; in the late-stage of osteoblast maturation the molecule is down-regulated [50]. Interestingly, the vector-mediated introduction of RunX2 in -/- animals strongly induces RANKL, inhibits OPG, and restores the differentiation of osteoclasts. In experimental animals, overexpression of RunX2 has inhibited the function of the canonical Wnt pathway through depletion of β-catenin, causing reduction in bone mass and bone volume. However, if overexpression of RunX2 is performed under lithium chloride treatment, which inhibits the β-catenin degrading enzyme GSK-3β, bone formation and trabecular bone volume are restored, osteoblasts are more differentiated than are bone-marrow derived osteoclast-like cells, and the RANKL/OPG ratio is reversed [51]. As a synthetic factor, β-catenin reactivation corrects the high bone resorption rate of RunX2 overexpressing mice, which is RANKL/OPG dependent.

## 7. Altered OPG-RANKL-RANK Signaling Combined with Abnormal Mechanical Load Leads to A Pro-Inflammatory Phenotype in Osteoarthritic Osteoblasts

In osteoarthritis, a close correlation exists between the dysfunction of the OPG/RANK/RANKL regulatory system and histological alterations of the subchondral bone, with the appearance of a pro-inflammatory osteoblast phenotype. Two different osteoblast types have been described: the so-called “low osteoarthritic osteoblasts”, which have low levels of PGE2, IL-6 and OPG and high levels of RANKL, and “high osteoarthritic osteoblasts,” which have increased expression of PGE2, IL-6, and OPG, along with decreased levels of RANKL (Figure 1) [52,53]. Ligand-binding activation of the ephrin B receptor suppresses RANKL along with pro-inflammatory cytokines IL-1 and IL-6 but not OPG. Sanchez et al. [54] developed an interesting model to study the gene and protein expression profile of subchondral osteoblasts. They presumed that abnormal mechanical pressure triggers dysregulated metabolic pathways in subchondral bone. They isolated and cultivated osteoblasts from sclerotic and non-sclerotic zones of the subchondral bone from osteoarthritis patients undergoing total knee replacement. After 28 days of cultivation to allow synthesis of pericellular matrix, mechanical shocks with 1 Hz frequency were applied for four hours to mimic mechanical loading. Under this condition, both sclerotic and non-sclerotic osteoblasts had increased amounts of inflammatory cytokines IL-6 and IL-8, Cox-2, the degradative metalloproteinases MMP-3, -9, -13 and RANKL, along with a significant decrease of OPG production. Non-sclerotic osteoblasts were more responsive to the pressure challenge than were sclerotic osteoblasts, but most of the differences were abolished after compression ceased [54].

## 8. Several Wnt Molecules Are Implicated in Subchondral Bone Sclerosis and Cartilage Degradation

Apparently presenting biologic and metabolic inertia, the micro-environmental system of osteochondral tissue has a unique dynamism, characterized by a continuous remodeling by various cellular participants and intra- and inter-cellular communication pathways [55]. Among the well-known cellular formations, subchondral osteoblasts have a remarkable role in both bone and joint homeostasis. Putative links and crosstalk between the underlying bone and articular cartilage cell lines, e.g., chondrocytes, have been in the limelight of current research as potential pharmaceutical targets in osteoarthritis [56,57]. Molecules of the canonical pathway, but also of the non-canonical pathway, are involved in the maintenance of bone homeostasis. Extensive studies and recent review publications regarding the role of Wnt signaling reveal an opposite dual effect of Wnts in the physiopathology of bone and joint diseases. Activation of these metabolic pathways are considered to contribute to the destruction of the articular cartilage by enhancing catabolic events or contributing to the loss of chondrocyte phenotype and being associated with chondrocyte dedifferentiation and extracellular matrix degradation [47,58,59,60,61]. In contrast, intact Wnt signal-transduction concurrently leads to subchondral bone remodeling, whilst both excessive and insufficient activation can induce osteoarthritic changes, e.g., osteophyte formation. Hence, alterations and imbalance in the activation or repression of Wnt-mediated osteo-articular homeostasis are important risk factors for the appearance and evolution of osteoarthritis [59].

Human genetic studies strongly support the role of Wnt signaling in the regulation of bone mineral density [62,63]. Human Wnt1 sequence mutations are associated with osteoporosis or osteogenesis imperfecta [64], while deletion of Lrp5 causes osteopenia [65]. Silencing of the most important inhibitors, e.g., soluble Frp1, SOST and DKK-1, increases the number of osteoblasts and their activity [66,67,68]. Tornero-Esteban et al. [69] reported altered Wnt signaling during in-vitro osteogenesis from mesenchymal stem cells of osteoarthritis patients. Under–expression of both canonical (Wnt2b, Wnt9a) and non-canonical (Wnt5a, Wnt5b) secreted Wnt ligands occurred by day 21. Moreover, the silencing of the Wnt cascade was observed in the case of membrane receptor (FZD3, FZD6) and intracellular modulators (dishevelled, Dsv1) down-regulation.

Participation of Wnt agonists, such as Wnt-induced signaling protein 1 (Wisp-1, CCN-4), in the pathogenesis of osteoarthritis was disclosed by Blom et al. [70] in mouse experiments. Collagenase-induced osteoarthritis cell cultures had a 2.8-fold increase in the expression of Wisp-1 compared with that in naïve mice by week 3 of intervention; a similar tendency of Wisp-1 expression in spontaneous osteoarthritis-developing STR/Ort mice was seen. These findings indicate that Wisp-1 might be a key regulator of cartilage loss in osteoarthritis and determinant of cartilage deterioration in vivo [70]. Supportive results were presented by Chou et al. [71]: a canonical Wnt ligand, such as Wnt1, was associated with osteoarthritis by facilitating alterations in bone structure. Several publications ascribe a pivotal role of Wisp-1 in osteoarthritis, being implicated in various events that induce joint pathogenesis [72], e.g., chondrocyte hypertrophy [73], osteophyte formation, and subchondral bone sclerosis [74,75,76]. Sanchez et al. [61] have reported that sclerotic subchondral bone has an over-expression of alkaline phosphatase activity, numerous pro-inflammatory (IL-6, IL-8) and anti-inflammatory cytokines (TGF-β), and structural proteins (osteopontin, osteocalcin and collagen type I). Another ligand, Wnt16, has an indispensable role in the prevention of both bone [77] and articular cartilage [23] damage. Nalesso et al. [78] demonstrated that Wnt16 expression, after injury of articular cartilage, resulted in a buffered activation of Wnt signaling, re-established its homeostatic functionality, and prevented the exacerbation of cartilage breakdown by exaggerated Wnt activation.

Martineau et al. analyzed the expression of Wnt5a and LGR4 and LGR5 receptors in primary subchondral osteoblast cell cultures obtained from the tibial plateaus of osteoarthritis patients [79]. They found that Wnt5a, a classical non-canonical Wnt pathway activator, has a five-fold over-expression in osteoarthritis osteoblasts. Interestingly, besides the increased amounts of Wnt5a, LGR5 receptors (but not LGR4 receptors) were up-regulated, as determined in mRNA analysis and protein quantification. Shi et al. [80] have reported that Wnt5a mRNA silencing in rat chondrocytes might be a therapeutic target in osteoarthritis treatment, as up-regulation of Wnt5a promotes the degradation of type II collagen IL-1β, contributing to the progression of osteoarthritis.

Although recent evidence accentuates the momentous role of Wnt signaling in both bone and joint homeostasis, and numerous studies suggest that the silencing of Wnts- and Wnt-related pathways are beneficial from an osteoarthritis point of view, due to the complexity of these communication systems, a balanced biologic activity is mandatory for an equitable microenvironment on the bone-joint border (Table 1).

## 9. Hypertrophic Chondrocytes and Subchondral Osteocytes Are Sources of Protective Wnt Inhibitors

The effect of Wnt inhibitors in osteoarthritis was evaluated by Wu et al. [81] in subchondral osteoblasts of the tibial plateau. Immunohistochemical and qRT-PCR analysis of β-catenin, TCF-4, and SOST/Wise revealed an elevated expression of β-catenin and TCF-4 in the intermediate and late stages of osteoarthritis, whereas sclerostin levels were lower compared to levels in the early-stage osteoarthritis samples. Evidence that metabolic interplay and interdependence of bone and joint are cross-linked by Wnt signaling is further supported by the development of osteoarthritis under treatment with anti-osteoporotic agents targeting Wnt signaling. Monoclonal antibodies targeting sclerostin e.g., blosozumab and romosozumab, which are inhibitory glycoproteins of Wnt signaling, contribute to the activation of the Wnt pathways, thereby inducing osteogenesis, improving bone strength, and stimulating osteoblast activity [82]. The injurious effects of sclerostin deletion on cartilage was described by Bouaziz et al. in SOST knock-out mice [83]. Their findings suggest that lack of sclerostin contributes to the growth of trabecular bone and increases the bone volume/tissue volume ratio but does not augment osteophyte formation. In contrast, more severe cartilage damage due to joint instability occurred in sclerostin knockout mice in comparison to that in wild-type mice that had undergone the same joint destabilizing operations. Wu et al. [84] recently reported that sclerostin secretion by chondrocytes is not consistent in the various stages of osteoarthritis: in early disease, sclerostin secretion by pre-hypertrophic chondrocytes is infinitesimal, suggesting that during the onset of osteoarthritis, sclerostin is secreted by subchondral osteocytes into the cartilage, whereas by mid-stage and late-stage disease, hypertrophic chondrocytes have progressively more sclerostin secretory properties. These observations are in concordance with those of Wu et al. [81] regarding sclerostin secretion by subchondral bone in the early stages of osteoarthritis.

In osteoblast cell cultures, down-regulation of the Wnt agonist R-spondin-2 gene expression is associated with increased levels of sclerostin [85,86]. Osteocytes communicate with osteoclasts and osteoblasts [87], exerting an anabolic effect by osteoblast-mediated osteogenesis via Wnt signaling [88,89,90]. Robling et al. [91] reported that, under the loading and unloading of mechanical stimuli of bone, sclerostin has a central role in the mechanotransduction cascade. Their results suggest that under mechanical loading, the expression of sclerostin is deficient, whereas the opposite occurs in bone unloading. This observation is evidence that osteocytes coordinate bone metabolism since they are the major sclerostin-secretory cells in bone. DKK-1-mediated inactivation of Wnt signaling down-regulated vascular endothelial growth factor (VEGF), resulting in decreased accretion of subchondral bone, inactivation of osteoblasts, and reduced formation of osteophytes. Funck-Brentano et al. [92] also investigated the inhibitory effect of DKK-1 on VEGF expression in meniscectomized Topgal mice. Their findings present significant VEGF under-expression, observed by both immunohistochemistry and gene expression analysis, in the presence of DKK-1. Wnt signal suppression by DKK-1 is also advantageous in osteoarthritis, as this communication path is activated under pathological conditions in articular cartilage.

## 10. Wnt Activation Causes A Metabolic Shift through the Mammalian Target of Rapamycin Complex

Osteoblasts rely primarily on glucose but somewhat on fatty acids and glutamine for their energy needs. Glucose uptake in osteoblasts is mediated by GLUT-1, -3, and -4 and seems to be a key regulator of early osteoblast commitment by stabilizing RunX2, an inducer of Wnt signaling; in osteoblast cultures, GLUT-1 blocks the ubiquitination of RunX2, thus prolonging its half-life. It has been reported that the anabolic functions of Wnt are linked to an increased aerobic glycolysis in osteoblasts [93]. Wnt proteins generate this effect via the activation of the mammalian target of rapamycin complex (mTORC) 2 complex, which in turn activates a few key enzymes in the glycolytic pathway: hexokinase 2, phosphofructokinase 1, lactate dehydrogenase A, and pyruvate dehydrogenase kinase 1. The main mechanism of fuel utilization to produce energy in osteoblasts is anaerobic glycolysis. This mechanism is not fully coupled to the otherwise active tricarboxylic acid cycle and oxidative phosphorylation, but rather it is shifted to form lactic acid since the oxidative decarboxylation of pyruvate is inhibited through pyruvate dehydrogenase kinase 1 [94]. This shift may be a result of hypoxia and might have the benefit of low-level reactive oxygen species.

Recent communications have aimed to clarify the crosstalk between Wnt signaling, with various pathways implied in modulating various cellular functions. Wnt proteins, such as Wnt 7b, featured in non-canonical signaling, characteristically activate the serine threonine kinase mTORC1, promoting protein synthesis and increase in bone mass [95,96,97].

Zeng et al. [98] have reported that mTORC signaling suppresses Wnt by down-regulating, via receptor internalization, Frizzled-receptor expression on the cell surface. Interestingly, Lin et al. [99] have disclosed that inactivation of mTORC1 in subchondral pre-osteoblasts prevented abnormal subchondral bone formation and sclerosis, subsequent cartilage degeneration, and eventual development of post-traumatic osteoarthritis. Furthermore, the activation of mTORC1, for example by abnormal mechanical loading, results in aberrant subchondral bone formation due to the proliferation of subchondral pre-osteoblasts and Cxcl12 secretion, contributing to the pathology of osteoarthritis. Thus, pharmacologic inhibition of mTORC1-related mechanisms might be a promising therapy of osteoarthritis [99].

## 11. Metabolic Disease and Osteoarthritis

There is growing evidence for a strong link between obesity, metabolic syndrome, and osteoarthritis [100,101]. High carbohydrate-high fat diet in rats favors osteoarthritis, local synovial inflammation with synovial thickening, and fibrosis, with increased amounts of IL-6 and IFN-γ, which counter the anti-inflammatory role of IL-10. Synovial fluid of rats fed high-carbohydrate diets triggers macrophage polarization to the iNOS-positive (inducible nitric-oxide synthase) M1 subtype. These cells, if co-cultivated with chondrocytes, down-regulate Sox-9 but enhance the expression of RunX2, MMP-13, ADAMTS5, COL10 [101]. It seems that high-calorie intake generates a low-grade pro-inflammatory state in the joints [100]. Some data indicate that high serum glucose increases cartilage breakdown through the activation of MMPs, which induces cellular senescence in chondrocytes [102]. The diabetes-associated subtype of osteoarthritis might be an appropriate model for understanding the complex relationships between metabolic disturbances and osteoarthritis.

The NHANES III study revealed that almost osteoarthritis patients have metabolic syndrome [103]. There could be many explanations for this linkage: 1) overweight and sedentary lifestyle are associated with abnormal and restricted mechanical load, along with local hypoxia and low-metabolism adaptation; 2) leptins synthesized by adipose tissue trigger catabolic responses in cartilage; and 3) the coincidence of atherosclerosis and osteoarthritis is higher in women [104]. We previously found slightly but non-significantly higher OPG levels in atherosclerotic females than in males [30]. Chondrocytes increase their glucose uptake via the glucose transporter 1 in response to hypoxia [105]. Important metabolic adaptation takes place in chondrocytes via 5-AMP-activated protein kinase (AMPK), a kinase that favors the synthesis of adenosine triphosphate in normal chondrocytes. AMPK is active in its phosphorylated form and activates phosphofructokinase-2, which in turn keeps phosphofructokinase-1 active. AMPK activation in osteoarthritis cartilage is decreased, and its inhibition enhances the catabolic responses to pro-inflammatory signals, such as IL-1β and TNFα [106]. Thus, hypoxia and low-grade inflammatory signals sustain high degrading-enzyme activities.

## 12. Metabolic Regulator PPARβ/δ Is A Putative Link between OPG/RANKL and Wnt

Peroxisome proliferator-activated receptor β/δ (PPARβ/δ) is a nuclear receptor isoform that possesses two transactivation domains, one DNA, and one ligand binding domain [107]. Through the usual activation, PPARβ/δ hetero-dimerizes with 9-cis retinoic acid receptor and binds to the PPAR element of various genes [108]. PPARβ/δ directly suppresses the MAPK signaling pathway, and it also inhibits the p65 subunit of NFκB and its binding to DNA [109]. Furthermore, the molecule is a reputed master regulator of energy metabolism, enhancing glycolysis but decreasing oxidative phosphorylation in mitochondria. This effect has proved beneficial in regenerative processes [107]. PPARβ/δ overexpression in high energy-demanding organs, such as the heart, stimulates glucose uptake via glucose transporter 4, and consequently its glycolytic degradation activates FOXO1 and the enzymes pyruvate PDK4 and lactate dehydrogenase B [110]. PPARβ/δ has strong anti-apoptotic, anti-fibrotic, anti-angiogenic, and anti-inflammatory effects, which inhibit iNOS, Cox-2, TNFα, and VEGF [108,109,111]. In an osteoblast model, PPARβ/δ significantly amplified Wnt3a and Wnt10b signaling, resulting in β-catenin accumulation and high-level TCF-mediated transcriptional activity. In this activity, two mechanisms were involved, one through transcriptional interaction with PPAR-responsive elements in the LRP5 promoter, and another via direct interaction with β-catenin. PPARβ/δ also induced, in a Wnt-dependent manner, OPG, osterix and RunX2. In osteoblast/osteoclast co-cultures, PPARβ/δ stimulates the expression of OPG, thereby decreasing the RANKL/OPG ratio. In mice, PPARβ/δ-deficient animals had numerous osteoclasts and osteopenia, whereas pharmacological activation restored the normal RANKL/OPG ratio and bone mineral density (Figure 1) [112]. However, less is known about the role of PPARβ/δ in osteoarthritic tissue. The highlighted interactions suggest that it is important in the homeostasis of subchondral bone.

Observations on the mutual regulation of expression between PPARβ/δ and homeostatic hypoxia inducible factor-1α (HIF-1α) also support this statement [113]. Some natural compounds, such as dioscin (a saponin), and mangiferine (a xanthone glucoside), have up-regulated another isoform, PPARγ in osteoarthritic cartilage and chondrocytes. Dioscin protects the cartilage by antioxidant, anti-apoptotic, and anti-inflammatory effects that down-regulate the canonical Wnt pathway, while mangiferine inhibits IL-1β-induced PGE2 release, canonical NFκB signaling, and activation of MMP-1, and MMP-3 [114,115]. In summary, PPAR molecules are protective for osteoblasts and interfere with multiple cellular pathways, implicating both members of the Wnt family and OPG. PPARβ/δ assures energy by enhancement of glycolysis but eliminates the formation of reactive oxygen species by down-regulation of the oxidative phosphorylation, along with strong anti-fibrotic, anti-inflammatory and pro-synthetic effects; PPARγ also seems to halt apoptosis and inflammation.

## 13. Macrophages Are Important Sources of Pro-Inflammatory Mediators in Subchondral Bone and Bone Marrow

Macrophages play an important role in osteoarthritis, being accumulated not only in the synovial tissue but also in the subchondral bone marrow. These cells are important sources of proinflammatory cytokines, and the ratio of CD86+ M1 type/CD163+ M2 type macrophages has been correlated with the Lawrence-Kellgren disease severity score in human osteoarthritis [116].

In tibial plates of patients who underwent total knee arthroplasty, single-cell suspension flow-cytometry revealed significant percentages of CD14+ macrophages and CD45+/HLA-DR+/CD115+ osteoclast progenitors [117]. Macrophages and osteoclasts are accumulated in the osteoarthritic subchondral bone, especially in the osteosclerotic regions [118]. M1-type cells are proinflammatory, secreting abundant quantities of IL-1, IL-6 and TNF-α; in contrast, the M2 subtype has reparatory functions [119]. According to recent suggestions, the accumulation of M1-type and failure of M1-to-M2 differentiation in the synovium might be the most important factor sustaining the low-grade inflammation of cartilage [119,120]. The M1-M2 transition promotes resolution of inflammation and may be beneficial in the ischemia-reperfusion-associated inflammatory injuries of various tissues [121,122,123]. M2-type macrophages can switch in some circumstances into fibroblast-like collagen-synthesizing cells [124]. In experimental circumstances, M2-type macrophages assist the osteogenic differentiation of mesenchymal stem cells. Macrophage polarization into the M2-type phenotype via LPS and IL-4 treatment promoted alkaline phosphatase activity and expression of RunX2, collagen I, and osteocalcin after 14 days of coculture with MSCs [125]. In some tissues, such as the kidney, canonical Wnt signaling pathways through β-catenin activation cause M2 polarization, excessive fibrosis, and aggravation of chronic kidney disease. Wnt3a, together with IL-4 and TGF-β, and Wnt5a, induce M2-type differentiation and alternative macrophage activation in the kidney [126].

Whereas M1-type synovial macrophages probably are important sources of proinflammatory mediators in the cartilage, both M1- and M2-type cells might influence remodeling and extracellular matrix mineralization in subchondral bone [127], and osteoclasts facilitate the formation of microchannels through the subchondral bone-cartilage interface [128].

## 14. Inflammatory Cytokines, Cartilage Matrix Degradation and Wnt

Inflammatory cytokines can diffuse into the cartilage matrix and induce local synthesis of MMPs. Locally secreted MMP-3, -9 exert catabolic functions on the extracellular matrix. In explanted mouse cartilage, cyclic mechanical compression of 1 MPa at 0.5 Hz for 2-24 h stimulated Cox-2 and PGE synthase in a time-dependent manner [129]. We previously reported the presence of low-grade inflammation in osteoarthritic rat cartilage identified with Cox-2 immunostaining [11]; Cox-2 has reportedly triggered many MMPs in various tissues [130,131,132,133]. Natural substances and endogenous inhibitors can suppress the activation of Wnt in chondrocytes. Polygalacic acid administration ameliorates the IL-1β-induced Cox-2 and consecutive MMP-1, -9 and -13 expression in rat chondrocytes in parallel with down-regulation of β-catenin nuclear translocation and MAPK signaling through ERK1/2 and JNK kinases [134]. Sclerostin, with osteoblasts and osteocytes, is overexpressed also in chondrocytes of focal cartilage lesions of surgically generated osteoarthritis, exhibiting complex effects: inhibition of aggrecanolysis, increased expression of structural components, such as collagen type II and aggrecan, downregulation of catabolic enzymes and their inhibitors, MMP, distintegrin, ADAMTs, and tissue inhibitors of metalloproteinases (Figure 2) [135].

Senescent chondrocytes are important sources of inflammatory cytokines in cartilage. These cells exhibit a senescence-associated secretory phenotype and can trigger an epithelial-mesenchymal transition-like phenotype switch through paracrine effects on surrounding cells [136]. One marker of aged chondrocytes is senescence-associated β-galactosidase. When transferred to healthy mice, the cells have also down-regulated TGF-β, abundantly synthesized and secreted MMP-1, -3, -9, -13, and provoked osteoarthritis-like lesions [137]. There is no evidence that Wnt signaling plays a direct role in senescence; however, mTORC might be a link, since rapamycin inhibits the translation of senescence-associated proteins, and cartilage-specific loss of mTORC increases AMPK, which may be beneficial in osteoarthritis [138]. Thus, Cox-2 mediated pathways of inflammation might be induced both by diffused mediators and senescent chondrocytes; natural and endogenous Wnt inhibitors seem to suspend MMP activity and the consequential cartilage deterioration.

## 15. Wnt Signaling and Chondrocyte Apoptosis

The contribution of chondrocyte apoptosis to the development of osteoarthritis is a controversial subject; some specialists claim that apoptosis is a central process in the disease pathology, and others attribute to it only 0.1% incidence [8]. The calcified cartilage layer is the only area where large gaps, as indicators of chondrocyte apoptosis, are evident. The paucity of chondrocytes in this cartilage zone does not seem to affect the joint under normal conditions, but it may be detrimental in advanced stages of the disease, when the calcified layer is enlarged. Dead cells are not effectively removed, and their products, such as pyrophosphate or precipitated calcium, may contribute to the pathological degradation of cartilage.

Studies of Zhu et al. [139] revealed the role of β–catenin signaling in cell apoptosis. The authors generated COL2A1-ICAT–transgenic mice, in which severe articular cartilage destruction and increased chondrocyte apoptosis developed. These results are evidence that the Wnt/ β-catenin signaling pathway has an anti-apoptotic function. However, TCF4, a downstream effector of Wnt/β–catenin signaling, induces MMP expression and apoptosis in human articular chondrocytes. Bin et al. [140,141] have reported that the pro-apoptotic action of TCF4 is not a consequence of activation of Wnt signaling but is due to the strong interaction with canonical NFκB signaling by activating IKK (inhibitor of κB kinase), thus contributing to the nuclear translocation of NFκB. This finding suggests that TCF4 is a potential pharmacological target in the treatment of osteoarthritis. Hwang et al. investigated the regulatory effects of Wnt7a, finding that it induced dedifferentiation of primary-culture articular chondrocytes and inhibited apoptosis. The authors concluded that Wnt-7a suppresses nitric oxide-induced apoptosis by activating cell-survival signals (including PI3-kinase and Akt activity), a finding that suggests that Wnt proteins exert anti-apoptotic effects acting through a mechanism that is independent of β-catenin. Other authors have suggest that Wnt proteins participate in cartilage degeneration by enhancing dedifferentiation, not by regulating apoptosis [142].

## 16. OPG and RANKL in Osteoarthritic Cartilage

OPG and RANKL are involved not only in the pathways that affect subchondral bone; both molecules are expressed also in osteoarthritic cartilage (Figure 2). RANKL has been found expressed by chondrocytes of the deep layer, while OPG is expressed especially in advanced disease [143]. An altered OPG/RANKL ratio was present also in osteoarthritic synovial fluid and chondrocytes, and OPG exerts positive effects, whereas RANKL is deleterious. OPG knockout mice develop severe degenerative multiple joint disease, form thinner cartilage layers, and have progressive loss of cartilage matrix [144]. These features are accompanied by low proliferative capacity, high apoptotic rate, and low collagen I and II but high collagen type X synthesis [143]. RANKL null mice have structural disorders of the cartilage growth plate along with disruption of the columnar organization of chondrocytes. RANKL is expressed by osteoarthritic chondrocytes, but in its soluble form, it probably does not participate in chondrocyte activation or cartilage deterioration, since exogenous RANKL does not trigger NFκB activation or transcription of pro-inflammatory cytokine genes. Upton et al. [143] have suggested that soluble RANKL diffuses from chondrocytes across calcified cartilage into the subchondral bone and regulates osteoblasts. However, RANKL more likely diffuses from the subchondral bone into the cartilage since osteoblasts and osteocytes are much more numerous than chondrocytes in the same tissue volume. Glucosaminoglycans of the matrix (heparin-, dermatan-, and chondroitin-sulphates) inhibit OPG-RANKL interaction. The OPG-RANKL complex is internalized either with syndecan-1 or via the clathrin-mediated pathway and are degraded inside the cell. Thus, OPG is an inducer of cellular RANKL degradation [53].

High mobility group box-1 (HMGB-1) is a conserved nuclear non-histone protein that plays a role in the fine tuning of inflammatory responses. HMGB-1 is an inducer of canonical NFκB signaling, which in its extracellular form is a strong “danger signal” and increases the expression of OPG in osteoblasts [145]. HMGB1 inhibition in IL-1β-triggered chondrocytes generates a dominant negative effect on a large set of inflammatory mediators, thereby down-regulating PGE2, Cox-2, iNOS, and MMP-1, -3 and -9 [146].

RANKL production leads to cartilage destruction, as increase in the RANKL/OPG ratio is associated with high MMP-13 synthesis in IL-1β-triggered SW1353 chondrosarcoma cells [147]. Low OPG and high RANKL have been associated also with increased expression of MMP-9 in a mixed osteoporosis-osteoarthritis model [148]. In a study investigating patients with knee osteoarthritis, the authors found increased mRNA levels of IL-6 and RANK associated with high expression of MMP-13 compared with levels in patients with femoral-neck fractures [149]. IL-1β at 10 ng/mL has been found to cause increased production of MMP-3, -13, ADAMTS-4, ADAMTS-5 and RANKL, whereas 100 μg/mL hyaluronic acid reversed this effect [150]. According to the observations mentioned above, a high RANKL/OPG ratio is linked to the up-regulation of catabolic effectors; the HMGB-1-mediated OPG induction might be a compensatory phenomenon.

## 17. Wnt Signaling, Chondrocyte Differentiation and Survival

Healthy chondrocytes are in cell-cycle arrest, whereas osteoarthritic chondrocytes have been found to have low proliferative activity. Proliferative activity in osteoarthritis can be explained partly by the fact that chondrocytes can access growth factors present in the synovial fluid [151]. Proliferation contributes to the formation of osteoarthritis-specific chondrocyte groups [8,151]. Canonical Wnt/β-catenin signaling is a regulator of differentiation of osteo-chondroprogenitors into either osteoblasts or chondrocytes. If Wnt signaling is repressed, the osteo-chondroprogenitor cells adopt a chondrogenic fate [152,153,154].

In adult cartilage, Wnt signaling is necessary for maintenance of cartilage homeostasis and the fully differentiated chondrocyte phenotype, which is characterized by prolonged cell survival and lack of differentiation towards hypertrophy [57,153]. Osteoarthritic chondrocytes resemble mesenchymal stem cells in many features [127]. In healthy conditions, the chondrocytes are in cell-cycle arrest or have low mitotic activity. In a model developed by Varela-Eirin et al. [155], an epithelial-to-mesenchymal transition, which is regulated by TGFβ, IKK-2/IκBα/NFκB and Wnt proteins, takes place in osteoarthritis [156]; resulting a fibrogenic (re-differentiated) phenotype with increased N-cadherin, vimentin, fibronectin, collagen and MMP levels. In the dedifferentiated intermediary stage, chondrocytes can proliferate. Both canonical and non-canonical Wnt agonists exert modulatory effects on the differentiation stage, synthetic capacity, and fate of chondrocytes.

Wnt5a expression in chondrocytes promotes a catabolic shift, triggering MMP-1, -3 and -13 expression [157]. XAV-939, a small molecule with inhibitory effect on Wnt signaling, reduces the proliferation rate of synovial fibroblasts and the synthesis of collagen type I without affecting chondrocyte proliferation but increasing COL2A1 formation [158]. Suppression of transcription factors or the β-catenin complex in chondrocytes may have major regulatory effects. NF-κB inhibition by siRNA-coupled nanoparticles ameliorates early inflammation in osteoarthritis generated by the mTORC1 and Wnt/β-catenin complex [98]. The never-in-mitosis gene-related kinase, NEK2, also seems to play an important role in the modulation of the cell cycle of chondrocytes: NEK2 suppression by miRNA or siRNA decreases β-catenin levels, β-catenin phosphorylation, MMP-13, Bax and p53, thereby inhibiting apoptosis while up-regulating collagen type II and aggrecan along with Bcl-2 production and proliferation [159]. Thus, Wnt agonists trigger the production of degradative enzymes in chondrocytes, while inhibitory molecules and endogenous antagonists increase the cartilage-specific collagen synthesis and maintain the proliferative capacity of chondrocytes.

## 18. Canonical and Non-Canonical Wnt Agonists are Negative Regulators of the Cartilage Extracellular Matrix

Abnormal Wnt signaling promotes progression of osteoarthritis by enhancing catabolic activity of chondrocytes or suppressing gene expression of extracellular matrix components. Stimulation of Wnt/β-catenin signaling leads to cartilage degradation through several mechanisms [139]. Increased activation of canonical Wnt pathways in osteoarthritic human cartilage has been reported, and studies conducted on animal models found a relationship between activation of the Wnt signaling and the osteoarthritis-like phenotype [57]. Wnt3a is negative regulator of cartilage extracellular matrix content. Treatment of chondrocyte cultures with Wnt3a leads to reduced matrix gene expression, loss of proteoglycans, and increased gene expression of proteases MMP-3, -13, ADAMTS-4 and -5 [160]. Wnt5a, through a non-canonical pathway, promotes the activation of catabolic signaling by increasing MMP-1, -3, -13 gene expression and MMP-1, -3 protein production in human articular chondrocytes. In addition, Wnt-5a suppresses aggrecan gene expression and, to a lesser extent, type II collagen expression [157]. In contrast, Wnt7a inhibits IL-1β-induced catabolic gene expression (MMP 1, MMP 13). Down-regulation of Wnt7a in human osteoarthritis cartilage and an inverse correlation between Wnt7a levels and catabolic gene expression in human articular chondrocytes in vitro has been reported [161]. Results of the same study disclosed that Wnt7a prevented articular cartilage degradation in mice.

Bouaziz et al. [162] have studied the modulation of Wnt/β-catenin signaling by HIF1α in the light of cartilage metabolism (proteoglycan release) and MMP-13 expression. The authors found that HIF1α inhibited the Wnt/β-catenin signaling pathway and reduced MMP-13 expression, leading to low catabolic activity in osteoarthritis.

Recent studies have revealed the impact of Wnt/β-catenin levels on chondrocyte mechanotransduction in relation to extracellular matrix content of the cartilage. Praxenthaler et al. [163] have studied the influence of extracellular matrix content on major signaling pathways, including Wnt/β-catenin, in bioengineered cartilage. They found that balanced Wnt activity establishes the biological response to loading through controlling the synthesis of proteoglycan. Suppression of Wnt levels by extra accumulation or with pharmacological antagonists resulted in anabolic activity of chondrocytes. These findings are in line with the role of proteoglycans in diminishing Wnt activity by interacting with Wnt-ligands; for example, heparin sulphate can interact with Wnt-ligands and inhibit their binding to Frizzled receptors. Low Wnt-activity can prevent progression of osteoarthritis in mechanically overloaded cartilage [163]. These observations highlight the possibility of a reciprocal homeostatic interaction between degradative Wnt agonists (like Wnt3a and -5a) and proteoglycan components of the extracellular matrix.

## 19. Wnt Signaling Triggers A Phenotype Shift in Chondrocytes

The elementary activity of Wnt is to maintain chondrocyte homeostasis and low proliferative capacity. However, excessive activation triggers hypertrophic degeneration and excessive production of MMPs [70]. Conditional activation of β-catenin in experimental animals leads to the development of an osteoarthritis-like phenotype [164,165]. In osteoarthritis, chondrocytes also suffer phenotypic changes. Hypertrophic degeneration has been proposed as a central element of the histological changes in osteoarthritis [166]. In vivo and in vitro studies have shown that several factors, such as retinoic acid, chemokine, and IL-1, exert phenotypic modulation of the cells; these factors stop expressing aggrecan and type II collagen, shifting to aberrant expression of collagens I, III and V [167,168]. Most data suggest that major phenotypic changes occur in the superficial cartilage area, where chondrocytes express abnormal non-specific genes for cartilage composition. The cells also produce enzyme sets capable of degrading the extracellular matrix as well as many of the cytokines involved in the activation of catabolic processes inside the cartilage [8,151].

The Wnt signaling pathway plays an essential role in the perpetuation of chondrocyte differentiation. Wnt3a and Wnt5a and 5b promote chondrogenic differentiation and inhibit the development of hypertrophy, whereas Wnt4 and 8 have opposite effects. Wnt11 up-regulates RunX2 and probably increases susceptibility to hypertrophy, whereas Wnt16 inhibits FRZB [165]. Ryu et al. [167] reported that accumulation and transcriptional activity of β-catenin causes phenotype loss of differentiated chondrocytes in rabbit articular cartilage. β-catenin is a negative regulator of differentiated chondrocyte phenotype, controlling the expression of c-Jun. Besides, accumulation of β-catenin leads to decline in the synthesis of extracellular matrix in the cartilage by enhancing TCF/LEF transcriptional activity.

The studies of Nalesso et al. [169] have demonstrated that Wnt3a can concomitantly activate the β-catenin– and the G protein–mediated Ca^2+^/CaMKII/pathways in adult human articular cartilage. Acting through β–catenin, Wnt3a induces chondrocyte proliferation. On the other hand, activation of the CaMKII pathway by Wnt3a promotes de-differentiation of articular chondrocytes and loss of the phenotype (loss of expression of the chondrocyte markers COL2A1, aggrecan, and Sox9). Wnt signaling becomes a selective pharmacological target by acting simultaneously on various pathways. A significant reduction in the highly sulphated GAGs occurred when adult human articular chondrocytes were treated with 100 ng/mL of Wnt3a; Wnt3a altered the extracellular matrix synthesis function of the chondrocytes [169].

Other research has focused on loss of chondrocyte phenotype via non-canonical Wnt signaling, involving the stimulation of the Fz-6/Dsv-2/SYND4/CaMKIIα/B-raf/ERK1/2 cascade. The work of Xie et al. [170] presents the mechanism of chondrocyte de-differentiation regulated by non-canonical Wnt and ERK1/2 signaling. Fz-6, SYND4, and B-raf are physiological regulators of chondrocyte phenotype, and they may be targets in the treatment of osteoarthritis. Hwang et al. [142] have studied the effect of Wnt7a on primary culture rabbit articular chondrocytes. Wnt7a treatment inhibited type II collagen expression and promoted type I collagen expression, which are signs of chondrocyte dedifferentiation. Inhibition of type II collagen expression is mediated by β-catenin-TCF/LEF complex. Thus, many Wnt agonists favor the hypertrophic degeneration of chondrocytes, while others exert an opposite effect or mediate de-differentiation, with the concomitant loss of type 2 collagen and aggrecan production.

## 20. Hypoxia and Wnt Signaling Interference in Chondrocytes

Hypoxia plays an important role in the maintenance of chondrocyte homeostasis (Figure 2). Bouaziz et al. [162] reported that HIF-1α is down-regulated in osteoarthritic mice, and chondrocyte-specific deletion of HIF-1α enhances the destruction of cartilage through inducing MMP-13 activation. HIF-1α seems to have homeostatic functions regulating Sox-9, which is the main transcription factor of chondrocytes; it promotes matrix synthesis while it down-regulates RunX2 and TCF-4/β-catenin interaction, suspending Wnt signals along with endochondral ossification. These effects are dependent on bone morphogenetic protein-2 [171]. However, an important counter-regulatory mechanism related to hypoxia also has been described [172]. In the presence of pro-inflammatory cytokines, such as IL-1β and TNFα, canonical NFκB signaling is induced, resulting in a different expression profile, with increased MMP-13, VEGF, and collagen type X. HIF-2α overexpression and crosstalk with β-catenin may end also in chondrocyte apoptosis and endochondral ossification [173]. Hypoxia activates HIF-1α, which, together with osterix, inhibits osteoblast growth. Applying siRNA-mediated inhibition against HIF-1α, Chen et al. [174] documented that HIF-1α up-regulates sclerostin on the transcriptional level by direct binding to and activation of the sclerostin promoter. Thus, hypoxia has an important dual role, ant its final effects may be based on the action of cooperating signals.

Adiponectin is not expressed in normal cartilage, but it has been identified by immunohistochemistry staining in osteoarthritic cartilage and was correlated with the presence of collagen type II and aggrecan and to the expression of MMP-13. Adiponectin is also present in the synovial fluid, where it probably maintains a high turnover of the extracellular matrix; in chondrocyte culture, it perpetuates PGE2 and MMP-13 formation [175].

## 21. Wnt Signaling As A Promising Therapeutic Target in Osteo-Articular Pathology

Besides well-known osteoarthritis drugs [176], current trends in pharmacological management of osteoarthritis have opened new therapeutic horizons: inter alia monoclonal antibodies or modulation of Wnt signaling. Although several monoclonal antibodies have been developed for the treatment of arthritis, these are mainly recommended for the management of rheumatoid arthritis, e.g., adalimumab, infliximab, sarilumab, and canakinumab, often targeting pro-inflammatory cytokines (ILs, TNFα). Two new molecules, CRB0017 and GSK 2394002, are enrolled in preclinical studies as anti-ADAMTS-5 monoclonal antibodies [177,178,179]. Otilimab, a new molecular entity that inhibits granulocyte macrophage colony stimulating factor, developed by GlaxoSmithKline, has been recommended for the treatment of both rheumatoid arthritis and osteoarthritis and was withdrawn for osteoarthritis before phase II clinical trials [180]. Focusing on Wnt signaling, Deshmukh et al. [181] compared the inhibitory effect of SM04690 on Wnt signaling to that of well-known Wnt pathway inhibitors, e.g., IWR-1 and ICG-001, in SW480 and bone-marrow-derived hMSC cell-culture studies. The results, based on gene expression analysis and protein quantitation, indicated a 150–500-fold greater inhibitory action of SM04690 over Wnt molecules compared with that of the encountered compounds [181]. Yazici et al. [182] have conducted phase I (24 weeks) and phase II (52 weeks) clinical trials with the Wnt-signaling inhibitor SM04690, with 61 and 455 subjects enrolled, respectively. The study populations included patients aged 50–75 years with Kellgren-Lawrence grades 2–3 in knee osteoarthritis. The patients received 0.03, 0.07 and 0.23 mg of SM04690 treatment as a single 2-mL ultrasound-guided intra-articular injection. The findings indicate that SM04690 might be a useful and well-tolerated disease-modifying drug for the treatment of knee osteoarthritis [182,183]. The phase 3 clinical trials of this agent are undergoing approval by the Food and Drug Administration [184]. Recent findings describe the beneficial effect of resveratrol on osteoarthritis osteoblasts by up-regulating the cWnt pathways. Abed et al. [185] have reported that resveratrol contributes to the increment of β-catenin expression in subchondral osteoblasts and might stimulate cWnt signaling induced by cWnt signal ligands, such as Wnt3a. In other work, Pelletier et al. [186] have reported a positive effect of diacerein in subchondral osteoblasts. They found that besides an increase in intracellular β-catenin levels, the drug was translocated into nuclei. Furthermore, the underlying mechanism of these positive changes in the expression of β-catenin have been attributed to the inhibitor of DKK1/2 in a dose-dependent manner. Zhong et al. [187] have described the anti-inflammatory and chondro-protective effects of artemisinin in IL-1β-induced rat chondrocyte cell cultures. Artemisinin treatment halted the up-regulation of the participants in Wnt signaling (β-catenin, Wnt5a and GSK-3β) and down-regulation of FRZB. The suppression of Wnt signaling by artemisinin was also seen in human-derived osteoarthritis chondrocytes.

## 22. Conclusions

The subchondral bone-cartilage border is a regulatory interface of joints, containing newly formed vessels, channels, and pores that permit the diffusion of molecules throughout the whole unit. Two important signaling systems, the OPG-RANKL-RANK and the Wnt, interplay in balancing the resorption and remodeling of subchondral bone. RunX2 is a differentiation factor that is mandatory for bone formation and links these two signaling systems. In experimental animals, RunX2 overexpression stimulates RANKL, depletes OPG and β-catenin along with reduction of bone mass and bone volume. Osteoarthritic osteoblasts have two different phenotypes: low-synthesizer cells produce little IL-6, PGE2, and OPG, whereas high-synthesizers secrete more of these mediators but low amounts of RANKL. Cyclic mechanical shocks, mimicking aberrant loading, also cause up-regulation of inflammatory mediators and degradative enzymes MMP-3, -9, and -13 in osteoblasts from sclerotic and non-sclerotic subchondral bone regions. Wnt signal transduction in subchondral osteoblasts generates intense remodeling, and some agonists, such as Wisp-1, also are involved in the deterioration of osteoarthritic cartilage and osteophyte formation, whereas others, such as Wnt16, seem to be protective. Increased mRNA and protein levels of canonical Wnt5a have been detected in osteoarthritic osteoblasts and in chondrocytes. Inhibitors of Wnt, e.g., sclerostin, appear to inhibit joint cartilage damage, since sclerostin knock-out mice have had severe histological injury. In the resting state, osteocytes are important sources of sclerostin, whereas under mechanical loading, their synthesis is deficient. Another inhibitor, DKK-1, not only prevents Wnt signaling, but it also decreases the expression of VEGF, osteoblast activation, and osteophyte formation. Both canonical and non-canonical Wnt agonists are negative regulators of the cartilage extracellular matrix; activation of Wnt3a and Wnt5a results in loss of proteoglycans and increased expression of MMP-3 and MMP-13. Wnt signaling also causes loss of the normal phenotype in chondrocytes since it decreases the expression of COL2A1, aggrecan, and SOX9, a central transcription factor necessary for cellular homeostasis. OPG is protective in the cartilage, while RANKL and a high RANKL/OPG ratio are associated with a pro-inflammatory state, leading to destruction of cartilage tissue via activation of MMP-13. An important putative regulatory link between the OPG/RANKL and Wnt signaling systems is PPARβ/δ, a nuclear receptor isoform that promotes glycolysis, increases OPG expression, and inhibits Wnt and MMPs in osteoblasts and cartilage.

Hypoxia plays a dual role in chondrocytes: it activates primarily HIF-1α, which promotes matrix synthesis, silencing RunX2 and Wnt. However, in the presence of proinflammatory cytokines, hypoxic conditions turn on HIF-2α, which enhances VEGF, MMP-13, and aberrant collagen expression with extracellular matrix deterioration.

Crosstalk and metabolic interaction at the articular–osseous border are incompletely understood, but this anatomic entity is of emerging interest and importance; it is strongly regulated by Wnt and OPG-RANKL-RANK, and it has potential therapeutic targets in the medical treatment of osteo-articular diseases.

## Figures and Tables

**Figure 1 ijms-20-04653-f001:**
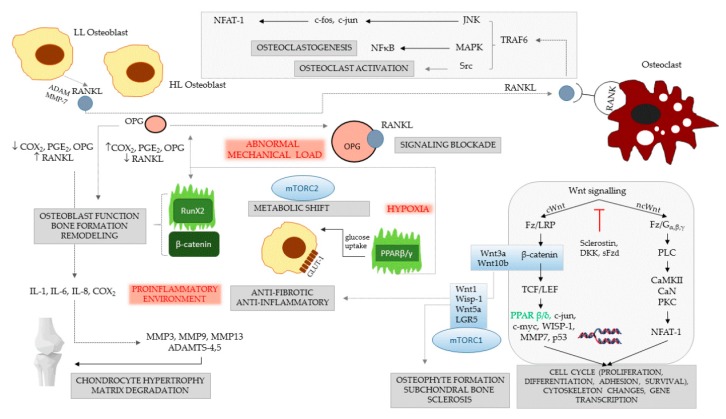
Signaling events and their consequences in osteoarthritic subchondral bone. Three classical factors—abnormal mechanical load, hypoxia and pro-inflammatory transformation play a trigger role in osteoarthritis (represented in red). Two types of osteoarthritic osteoblasts are illustrated: high-level (HL) and low-level (LL) synthesizer cells. The first type produces high levels of Cox-2, PGE2 and OPG and less RANKL; the second type has an opposite secretory pattern. The HL synthesizers maintain pro-inflammatory conditions in the subchondral bone, and they are increased in the presence of abnormal mechanical load, when they produce OPG, which may enhance focal osteosclerosis. The LL synthesizers release important quantities of RANKL, which activate osteoclasts by linking RANK. Signal cascades are activated via TRAF6, JNK, MAP and Src kinases, resulting in osteoclast differentiation and activation. Activation of Cox-2 and PGE2 stimulates the synthesis of IL-1, IL-6, and IL-8, which turn on the production of degradative MMPs in the cartilage. In normal conditions, osteoblasts rely for energy mainly on glucose, and PPARβ/γ enhances glucose uptake via the GLUT-1 transporter; however, in hypoxic conditions, the cells undergo a strong metabolic shift to anaerobic glycolysis, driven by mTORC2 that stabilizes RunX2. PPARβ/γ cooperates with canonical signal molecules, e.g., Wnt3a and Wnt10b, to exert anti-fibrotic and anti-inflammatory effects, further inducing OPG and RunX2 and fine-tuning osteoblast function, bone formation and remodeling. RunX2 alone induces RANKL and controls osteoclast function, but when acting together with β-catenin, it stimulates mainly osteoblasts, which increase trabecular bone volume. The two linker factors of OPG/RANKL and Wnt signaling, RunX2 and PPARβ/γ, are shown in green.

**Figure 2 ijms-20-04653-f002:**
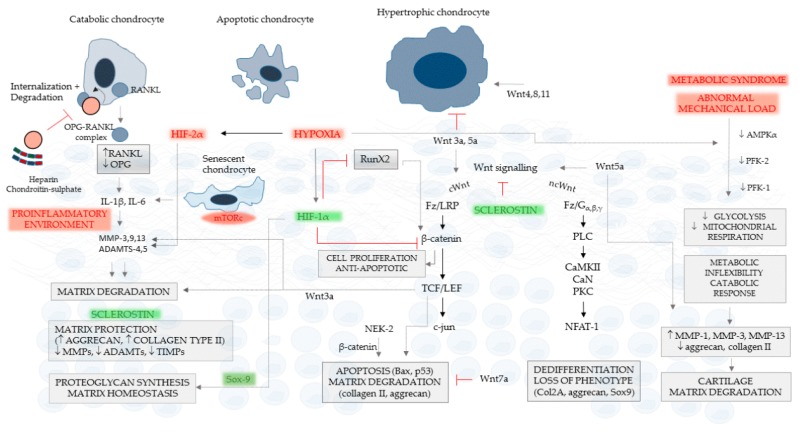
Effects of OPG/RANKL and Wnt signaling on chondrocytes and cartilage. The presence of catabolic, apoptotic and hypertrophic chondrocytes is a characteristic of osteoarthritis. Chondrocytes expressing RANKL release it in the soluble form, triggering IL-1β and IL-6 overexpression in the cartilage. Senescent chondrocytes contribute to this pro-inflammatory environment, increasing the synthesis of MMP-3, -9 -13, ADAMTS-4 and -5. The pro-inflammatory activity results in degradation of cartilage extracellular matrix. OPG links to RANKL and facilitates its internalization with consequent degradation. Sclerostin is a strong inhibitor of cartilage degradation, which suspends MMPs, ADAMTs and even their inhibitors—tissue inhibitors of metalloproteinases—stimulating in parallel aggrecan and collagen type II production. Canonical Wnt signaling and β-catenin are anti-apoptotic and stimulate chondrocyte proliferation; however, TCF/LEF is a downstream activator of apoptosis and matrix degradation. Hypoxia, triggering HIF-1α, generates an inhibitory effect on β-catenin and RunX2 and enhances the homeostatic effects of Sox-9. Non-canonical Wnt activation has different pathological consequences: chondrocyte dedifferentiation and loss of phenotype, the agonist Wnt5a up-regulating MMP-1, -3 and -13, and inhibition of aggrecan along with collagen type II. Wnt3a, together with Wnt5a, suspends hypertrophic transformation, while Wnt4, 8 and 11 are stimulatory. The metabolic syndrome and abnormal mechanical load down-regulate AMPKα, glycolysis and mitochondrial respiration in an adaptive response to hypoxia, which also results in a catabolic phenotype switch of chondrocytes. Factors of chondrocyte and cartilage deterioration are shown in red, and protective mediators are drawn in green.

**Table 1 ijms-20-04653-t001:** Role of Wnt signaling pathway in subchondral bone and cartilage.

Wnt Agonist/Antagonist	Subchondral Bone, Osteoblast	Cartilage, Chondrocyte
**Agonists/Inducers of Wnt Signaling**		
Wnt-1 (canonical)	Alteration in bone structure	
Wnt-3a (canonical)	-	Chondrocyte proliferation
Wnt-3a (non-canonical)	Activates mTORC2, Stimulates glucose consumption and lactate production of OBs	Reduces matrix gene expression Loss of proteoglycans Increases expression of MMP-3, -13 Loss of chondrocyte phenotype Altered ECM synthesis
Wnt-4	-	Chondrocyte hypertrophy
Wnt-5a (non canonical)	Generates osteoarthritic OB phenotype	Degradation of type II collagen Triggers MMP-1, -3, -13 expression Suppresses aggrecan gene expression, type II collagen expression
Wnt -7a (non-canonical)	-	Chondrocyte dedifferentiation Inhibition of apoptosis Inhibition of catabolic gene expression (MMP-1, -13) Prevents cartilage degradation
Wnt-7a (canonical)		Inhibition of type II collagen expression
Wnt-7b (non-canonical)	Bone mass gain Activates mTORC2 Stimulates glucose consumption and lactate production of OBs	-
Wnt-8	-	Chondrocyte hypertrophy
Wnt-11	-	Increased susceptibility for hypertrophy Upregulation of RunX2
Wnt-16	Prevents bone damage, Wnt signal buffering	Prevents cartilage damage
Wnt-10b	Stimulates glucose consumption and lactate production of OBs	-
Wisp-1	Subchondral bone sclerosis	-
Resveratrol *, diacerein *—upregulate Wnt signaling	Beneficial effect on osteoblasts	-
**Antagonists/Inhibitors of Wnt Signaling**		
DKK-1 *	Reduced accretion of subchondral bone Reduced osteophyte formation Inactivation of osteoblasts Control of OB proliferation	-
Sclerostin *	Control of OB proliferation	Inhibits cartilage damage
XAV-939 *	-	Reduced synthesis of collagen type I Increased COL2A1 formation
SM04690 *		Potential disease modifying osteoarthritis drug

*- not members of the Wnt family.

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
