# Peer review of "Regulatory Effects and Interactions of the Wnt and OPG-RANKL-RANK Signaling at the Bone-Cartilage Interface in Osteoarthritis"

_ijms, 2019, doi:10.3390/ijms20184653_

Round 1

Reviewer 1 Report

ijms-583021

Regulatory effects and interactions of the Wnt and OPG-RANKL-RANK signalling at the bone-cartilage interface in osteoarthritis

This review article summarizes known facts concerning the interplay between Wnt and OPG-RANKL-RANK signalling in osteoarthritis with a focus on the effects at the bone-cartilage interface. The general functional aspects of these regulatory systems and their impact on a variety of pathophysiological events (e.g., inflammation, sclerosis, matrix and cartilage degradation, alterations in chondrocyte biology, etc.) are discussed in detail.

Text and figures appear predominantly straightforward and clear. This review article covers the selected topic in detail, reflects relevant parts of the latest literature, and provides a broad and informative overview for the reader. Therefore, only a few a few minor points should be addressed.

1. The text of the manuscript has to be slightly revised due to a few missing and a number of redundant spaces as well as style inconsistencies (e.g., MTORC vs. mTORC). 2. All abbreviations should be defined in the text. Afterwards, only the abbreviations should be used throughout the text. 3. The comma setting should be checked and standardized (especially for enumerations). 4. The figures and the table should be linked to the text. 5. In general, the chapters contain large amounts of information. Thus, a summarizing sentence the end of relevant chapters (esp., 7, 11, 13, and 15-18) providing the respective key message would facilitate the understanding for the reader. 6. In chapter 5, (page 4, lines 174-175), it should be clearly distinguished between kinases and transcription factors. Which MAPK are induced by RANK activation? Moreover, the paragraph focusing on Runx2 in osteoarthritic osteoblasts should be shortened due to redundant information (e.g., concerning osteocalcin levels, alkaline phosphatase activity, and the role of Runx2 for bone formation). 7. Concerning chapters 5 (page 4, line 174), 11 (page 8, line 383), 14 (page 11, line 473), 15 (page 11, line 503), 16 (page 12, line 538), and 19 (page 14, line 627): please specify which NF-kappaB subunits are involved or which pathway (canonical vs. non-canonical NF-kappaB signalling) is affected. 8. Chapter 8 (page 7, lines 298-299): Is VEGF downregulated at the mRNA or the protein level (or both) by Wnt signalling? 9. In chapter 19 (page 10, lines 434-448), the information concerning pro-inflammatory and pro-destructive events is presented in a disorderly manner. Please improve. 10. In Chapter 19 (page 14), the paragraphs focusing on HIF-1alpha should be condensed and merged to one paragraph. 11. In Table 1, it should be distinguished between genes/proteins (e.g., members of the Wnt family) and other compounds (such as inhibitors).

Author Response

Dear Reviewer,

The Authors would like to express their gratitude for your prompt reply, thorough review and constructive remarks, which are useful for improving the quality of the manuscript.

We agree with the Reviewer’s comments and suggestions, so we prepared our revised version of the manuscript according to the recommendations. All the requested changes are comprised in the manuscript. In the response letter remarks, questions, etc. are in italics; our responses, corrections, etc. are in normal format.

The text of the manuscript has to be slightly revised due to a few missing and a number of redundant spaces as well as style inconsistencies (e.g., MTORC vs. mTORC).

The authors are thankful for the thorough revision of the format aspects of the manuscript. Spelling mistakes, other typographical errors and style inconsistencies were corrected throughout the text.

All abbreviations should be defined in the text. Afterwards, only the abbreviations should be used throughout the text.

Abbreviations were revised, completed and used in accordance with Reviewer’s requirements, as follows:

- all abbreviations were clearly defined after the first appearance in the manuscript

- afterwards, only the abbreviated form is used if it does not influence the context of the sentence

Alternatively, a list of abbreviations is provided in order to avoid ambiguous denotations of the abbreviations used throughout the manuscript and to avoid overloading the text with explanatory parenthesis, if appropriate.

The comma setting should be checked and standardized (especially for enumerations).

Comma settings were revised; redundant use of this glyph was corrected and standardized when used in enumerations.

The figures and the table should be linked to the text.

Figures and Tables have been linked to the text.

In general, the chapters contain large amounts of information. Thus, a summarizing sentence the end of relevant chapters (esp., 7, 11, 13, and 15-18) providing the respective key message would facilitate the understanding for the reader.

In order to improve readability and understanding, summarizing sentences have been introduced at the end of Chapters 7, 11, 13, 15, 16, 17 and 18.

In chapter 5, (page 4, lines 174-175), it should be clearly distinguished between kinases and transcription factors. Which MAPK are induced by RANK activation? Moreover, the paragraph focusing on Runx2 in osteoarthritic osteoblasts should be shortened due to redundant information (e.g., concerning osteocalcin levels, alkaline phosphatase activity, and the role of Runx2 for bone formation).

In chapter 5, we have explained that MAPK are induced by RANK; supportive information was introduced by including additional literature data.

The paragraph focusing on RunX2 was reconsidered, redundant information was deleted, and the text now only contains the essential information regarding the role of RunX2 in osteoarthritic osteoblasts.

Concerning chapters 5 (page 4, line 174), 11 (page 8, line 383), 14 (page 11, line 473), 15 (page 11, line 503), 16 (page 12, line 538), and 19 (page 14, line 627): please specify which NF-kappaB subunits are involved or which pathway (canonical vs. non-canonical NF-kappaB signalling) is affected.

The text of the manuscript was completed with the required elucidation of the activated NF-κB pathways under various conditions.

Chapter 8 (page 7, lines 298-299): Is VEGF downregulated at the mRNA or the protein level (or both) by Wnt signalling?

Under Wnt-inhibitory conditions, e.g. the presence of DKK-1, VEGF is under-expressed at both the protein level and mRNA synthesis. Supportive information was included in the manuscript.

In chapter 19 (page 10, lines 434-448), the information concerning pro-inflammatory and pro-destructive events is presented in a disorderly manner. Please improve.

Chapter 13 and 19 was reconsidered and modified in order to improve the legibility of these sections.

In Chapter 19 (page 14), the paragraphs focusing on HIF-1alpha should be condensed and merged to one paragraph.

Chapter 19, focusing on HIF-1α, was modified as requested by the Reviewer.

In Table 1, it should be distinguished between genes/proteins (e.g., members of the Wnt family) and other compounds (such as inhibitors).

Table 1 was reconsidered and modified in order to give a better transparency over the participants in Wnt signalling.

Reviewer 2 Report

This paper by Kovacs et al. represents an extensive review focusing on the interplay of two important signaling pathways, OPG-RANKL-RANK and Wnt, involving different cell types at the subchondral bone-cartilage border. The topic is complex, but of emerging interest and importance in relation to osteoarthritis. Altogether, despite the intricacies of the matter, the review is interesting, well organized and comprehensive.

Minor points:

Page 7, chapter 9: the features of metabolism in osteoblasts and metabolic shift by Wnt activation may be more clearly and orderly described.    

Pages 9, 14, 15: Reference of table and figures may be included in the text (Tab. 1, Fig. 1, Fig. 2).

Author Response

Dear Reviewer,

The Authors would like to express their gratitude for your prompt reply, thorough review and constructive remarks, which are useful for improving the quality of the manuscript.

We agree with the Reviewer’s comments and suggestions, so we prepared our revised version of the manuscript according to the recommendations. All the requested changes are comprised in the manuscript. In the response letter remarks, questions, etc. are in italics; our responses, corrections, etc. are in normal format.

Page 7, chapter 9: the features of metabolism in osteoblasts and metabolic shift by Wnt activation may be more clearly and orderly described.

Chapter 9 has been revised and minor modifications were made in order to improve the legibility of the section.

Pages 9, 14, 15: Reference of table and figures may be included in the text (Tab. 1, Fig. 1, Fig. 2).

References for Figures 1 and 2 and Table I have been included in the text.